# Multifunctional Geotextiles Produced from Reclaimed Fibres and Their Role in Ecological Engineering

**DOI:** 10.3390/ma15227957

**Published:** 2022-11-10

**Authors:** Damian Chmura, Anna Salachna, Jan Broda, Katarzyna Kobiela-Mendrek, Andrzej Gawłowski, Monika Rom

**Affiliations:** Faculty of Materials, Civil and Environmental Engineering, University of Bielsko-Biala, Willowa 2, 43-309 Bielsko-Biala, Poland

**Keywords:** waste, textiles, recycling, greening, wool

## Abstract

Earthworks in the vicinity of roads, open mines, subsidence tanks and other man-made objects can lead to the creation of slopes that undergo erosion. One of the methods that can prevent their degradation and reclaim them is the use of geotextiles. An environmentally friendly option is using geotextiles that are produced from reclaimed fibres. The purpose of this study was to examine the role of the mechanical and chemical properties of geotextiles, namely, ropes and fibres (containing wool and polypropylene), not only on the rate of the greening of slopes but also on the species composition of vegetation. We studied the floristic composition, species diversity, species growth and soil properties of four sites of reclaimed slopes on which 46 study plots (5 m × 5 m) were laid out. We found that some species were more confined to a higher content of wool and that other species were more confined to the content of polypropylene. Both materials caused a decrease in the Shannon–Wiener diversity but an increase in evenness under the impact of ropes when compared to the control. They both also contributed to a higher mean height of the plants when compared to the control. The rate of the plant colonisation process was markedly improved by the reclaimed geotextiles. A longer and more detailed study is required to examine the effect of geotextile ropes on habitat creation.

## 1. Introduction

Due to population growth, overall improvements in living standards and the shortening of lifecycles, the global production of textiles has steadily increased. The growing production has resulted in a higher amount of waste that is generated during both the production and consumption of textiles [1,2]. Each year, a huge amount of fibrous textile waste is discarded into landfills without receiving any treatment. The storage of textile waste raises numerous global problems, and for economic and environmental reasons, the recycling of textiles is urgently required.

Textile waste includes post-industrial, pre-consumer and post-consumer waste. Post-industrial waste consists of the remnants of fibres, yarns and fabrics that are generated at the various manufacturing stages of textiles, as well as the scraps of textiles that are generated by the clothing industry. This waste includes short fibres; fabrics that are rejected because of weaving, knitting or dyeing faults; garment-cutting waste; and the end lots from surplus production. Post-industrial textile waste consists of virgin or clean waste because the materials are discarded without being used. This waste is less diverse and can be relatively easily recycled to produce valuable products. Pre-consumer waste consists of rejected materials from the production of yarns and textiles that are discarded before they are ready for consumer use. This waste includes mill ends, scraps, clippings and goods that are damaged during production. Pre-consumer waste, which is known as “clean waste”, can easily be remade into different products of a similar quality and of a minimally lower value. The third group, post-consumer waste, refers to textiles that are discarded by consumers because they are no longer used due to damage, wear and tear; being out of fashion; or any other reason. This waste is separated from municipal waste or collected under various separate waste collection programmes. This waste, which is called “dirty waste”, is often mixed with other household items and contains impurities; dyes; and other materials, such as labels, buttons and zippers. Unlike industrial and pre-consumer waste, this waste is strongly diversified, contains products that were produced from both natural and man-made fibres and are of different colours. The recycling of post-consumer waste is much more complicated. In this case, closed-loop recycling that leads to obtaining more or less identical products is practically impossible [3]. This waste is only suitable for open-loop recycling in order to obtain a different product that is of a lower value and that is designed for other purposes or to fulfil other functions [4].

Textile waste is recycled using mechanical, chemical, enzymatic or thermal methods [5,6,7,8,9]. Due to its relatively simple procedure and low-cost disassembling of textiles, the mechanical method has gained significant importance. In this process, textile waste is mechanically shredded and disintegrated into loose fibres. For this process, a garnett machine, which consists of rotating drums with metal pins that destroy the textile structure, is used [10]. To obtain reclaimed fibres that are suitable for yarn spinning, an accurate sorting of the waste before the disintegration process is required. Because sorting post-consumer waste is a manual, time-consuming and costly process, mechanical recycling is commonly only used to roughly sort the waste. As a result, a mixture of different fibres of various colours of a much lower quality, which are not suitable for yarn spinning, is obtained. Because of its large variety and lower quality, this mixture can only be successfully used to produce different kinds of nonwoven fabrics [11].

In the process, reclaimed fibres are carded in order to form a fibrous web, which is later bonded using mechanical, chemical or thermal methods. Nonwovens that are manufactured from recycled fibres are commonly used as thermal and acoustic insulation materials for use in construction [12,13,14], filter materials [15,16], oil sorbents [17,18] and geotextiles [19,20,21].

Geotextiles are often used in various environmental engineering projects as erosion control materials and materials for soil reinforcement, slope stabilisation, post-industrial area reclamation, revegetation and groundwater contamination prevention [22]. Products that are designed for erosion control include mats or blankets, which are installed in close contact with the ground surface. The mats completely cover the protected surface and shield it from the impact of falling raindrops. Simultaneously, the mats reduce the biologically active surface, which can delay the development of protective vegetation.

As an alternative to conventional anti-erosion mats, innovative products created from nonwovens that are rolled into thick ropes were invented. The Kemafil technique developed in Germany in the 1970s is used to produce these ropes. This technique involves using a small circular knitting machine that is equipped with four hooked loopers that are arranged around a guiding tube. The threads that are guided by the loopers form a knitted sheath around the rope core [23,24].

The Kemafil technique has been widely used for manufacturing ropes that are produced from recycled textiles and other different materials for many years. The wide range of products includes sealing and insulating ropes that are used in the construction industry, drainage and irrigation ropes that are used in agriculture and horticulture, filling materials that are used in the production of upholstery products and ropes that are used for protecting goods during transport and for protecting slopes against erosion.

In the initial tests of anti-erosion applications, ropes with a diameter of a few centimetres were spread diagonally on a slope to form regular squares, and then they were fastened to the ground with metal anchors. Following the installation, the formed grid was covered with soil and then sown with grass seed. In further tests, thicker ropes, which were meandrically arranged on the slope and connected in segments with additional linking chains, were used. The segments were successfully used to protect an artificial embankment that serves as a noise barrier beside the road around the centre of Chemnitz (Germany). In further trials, the ropes were used to secure the unstable slopes of a disused lignite open mine and an abandoned gravel pit [24,25], drainage and road ditches [26,27,28], road embankments [29] and other slopes that were endangered by water erosion [30].

The ropes that are installed on slopes perform various functions. Similar to the brush, wattles or fences that are commonly used in bioengineering systems, the ropes form small retaining walls that divide the slope length into smaller segments. The ropes have sufficient strength to restrain the loads that are induced by soil weight and to protect the slope against gravitational soil movements [31]. Simultaneously, the ropes reduce the probability of slope failure and protect slopes against the shallow landslides that could occur by exceeding the mechanical resistance of the slope, which often results from a decrease in soil cohesion due to the increased pore water pressure that occurs via infiltration from precipitation events or snowmelt. Simultaneously, the ropes that are installed across slopes form mechanical barriers that retain the loose soil particles that are detached from the soil surface by raindrops, reducing they’re being washed out and transported down the slope. The ropes also restrain the formation of erosive rills or grooves on the surface of a slope and reduce the negative consequences of water erosion. A system of ropes protects slopes immediately after installation, which is of great importance for freshly profiled slopes without any protective vegetation.

In addition to their mechanical action, ropes play an important role in water management on a slope. During precipitation, the ropes form a system of micro-dams that form small cascades that slow down the stream of water flowing on the slope surface. Ropes that are made from a mixture of hydrophilic and hydrophobic fibres have a high water retention capacity and good capillary properties. Ropes that are installed on a slope act as a bio-wicking system with a dual function and serve both as a pump that sucks water from the soil and as a pipe that transports it down to the base of the slope. During heavy rains, the ropes absorb the excess water, reduce the amount of water flowing down the slope and eliminate any extreme runoff. Because of the capillary action of the ropes, the moisture is effectively and uniformly distributed on the entire surface of the slope. The ropes retain and temporarily store water and then gradually release it into the soil. In this way, the moisture is maintained for a longer time, even during longer periods of droughts. Consequently, the need for frequent irrigation of a slope is eliminated, and the negative impact of dry days on the development of vegetation is considerably reduced.

Previous investigations have shown that ropes accelerate slope greening, as well as the development of protective vegetation [32,33,34]. Ropes ensure favourable conditions for the development of vegetation in both the stage of seed germination and during seedling growth. Ropes sufficiently support vegetation development, even in difficult terrains where plant adaptation is strongly inhibited because of a poor substrate, a substantial land slope, intensive insolation or other reasons.

In the present study, the role of ropes in slope greening and vegetation establishment towards improved stability and long-term management of system stability in order to maintain a functional vegetation cover was analysed. In detail, the impact of ropes on three aspects of vegetation, namely, species composition, species diversity and vegetation growth, studied. We focused on the chemical nature of the material (synthetic material vs. wool) and mechanical structure (Kemafil technology vs. fibres). Wool and synthetic materials can contribute to higher soil moisture; in addition, wool, due to its decay, can cause a higher content of nitrogen in the soil.

We hypothesise the following:The influence of geotextiles manifests itself mainly in the influence on the growth of vegetation rather than on the differentiation of species compositionThe chemical nature of the material has a stronger effect on the species composition of the vegetation than its mechanical structure, i.e., ropes vs. fibres.The species composition is mainly dependent on soil parameters and the vegetation in the neighbourhood, and it is less dependent on the installation of geotextiles (ropes and fibres).Both added material types, synthetic material and wool, enhance the growth of vegetation in terms of the mean height of the plants; however, the influence of wool is stronger due to its properties.

## 2. Materials and Methods

### 2.1. Materials

In order to protect slopes, 100 mm-diameter ropes made from textile waste were used. The ropes were manufactured from strips of a needle-punched wool nonwoven or a stitch-bonded nonwoven produced from a mixture of mainly synthetic recycled fibres that had been obtained by shredding post-consumer textile waste using Kemafil technology. The ropes were wrapped with a sheath made of polypropylene twine with a linear density of 240 tex.

Before the installation on lopes, the segments of the ropes, which were arranged in a meander-like pattern, were prepared. The segments were 2 m wide, and the length was adjusted to the length of the slope. In the segments, the successive turns were stabilised using additional links made of polypropylene twine with a linear density of 10 g/m. During the installation, the segments were anchored to the top of the slope and fastened to its surface with steel “U-shaped” pins (Figure 1). Finally, the segments of the ropes were covered with a layer of topsoil.

The ropes were installed to protect four unstable slopes, located in the Silesia region in southern Poland, that had been exposed to intense water erosion. In two places, the ropes were covered with topsoil. In two other places, the ropes were covered with soil mixed with wool, polypropylene or polyester fibres. A list of the protected objects is presented in Table 1. The condition of these objects before the installation of the ropes is shown in Figure 2. More details on the location of these objects and their characteristics were presented in previous publications [25,26,27,28,29,30,31,32,33,34,35].

At three sites, Miedzyrzecze, Nieboczowy and Lipnik, the ropes were installed in spring at the beginning of the vegetation season. At the first two sites, the work was completed by levelling the surface layer. At the third site (Lipnik), immediately after the ropes were installed and the topsoil layer was levelled, grass seeds were sown on the slope. At the fourth site (Wapienica), the work associated with securing the slope was performed in autumn at the end of the vegetation season. Then, in spring, the slope was hydroseeded with a multi-component mixture of the seeds of various grass species along with several additional additives. The same mixture of seeds was used at each site.

### 2.2. Methods

#### 2.2.1. The Study Design and Botanical Studies

Botanical studies were conducted in 5 m × 5 m study plots that were laid out on each slope. The number of study plots varied from 5 to 23 depending on the local conditions. For the analyses, the plots that were protected with ropes made from the nonwoven wool or that had been covered with soil mixed with wool fibres were treated as “wool”. The other plots that were protected with ropes made from recycled fibres or that had been covered with soil mixed with polyester or polypropylene fibres were treated as synthetic material. In addition to the “wool”- and “synthetic”-type plots, at each site, control plots without ropes or fibres were laid out. The specific number of study plots on each slope is shown in Table 2.

On each study plot, phytosociological relevés were created using the Braun-Blanquet method [36]. The cover abundance of the vascular plants that were present was estimated visually. For the statistical analysis, the original Braun-Blanquet cover values (r, +, 1,2,3,4,5) were transformed into the medians of the percentage ranges, i.e., 0.1, 0.5, 5, 17.5, 37.5, 62.5, and 87.5. In a further step, biodiversity indices, namely, species richness (S), the Shannon–Wiener index, evenness (E) and the Simpson dominance index (D), were calculated. The height of the herbal layer at three randomly selected points was measured using a tape on each plot, and then the arithmetic mean value was calculated. The botanical studies were conducted during the peak of the vegetation season (June–July) two years after the ropes had been installed.

#### 2.2.2. Soil Studies

A mixed composite soil sample was taken from each study plot (four subsamples from the corner and one from the middle of a study plot). For each sample, the pH, the content of organic matter (humus—% of dry mass) and the content of organic carbon (%) were determined. Additionally, the content of nutrients, namely, phosphorus—P (P_2_O_5_—mg/100 g), potassium—K (K_2_O—mg/100 g), magnesium Mg—(Mg—mg/100 g), nitrates (N-NO_3_—mg/kg of dry mass) and ammonia ions (N-NH_4_—mg/kg of dry mass), were measured. The measurements were taken in accordance with the standard procedure of the Chemical and Agricultural Station research station in Gliwice [37].

#### 2.2.3. Data Analysis

To analyse the impact of the protection systems that were used for vegetation development on the slopes, the presence of ropes, the type of raw material that was used for their production and the type of fibres used to reinforce the soil were taken into account. The analysis included three variables, namely, the ropes, synthetic material and wool, which were treated as dummy variables, i.e., for the presence or absence of data, which were coded as 0 or 1, respectively.

All of the statistical analyses were conducted in the R language and environment [38]. The accepted level of significance was *p* < 0.05. For the ordination analyses, the “vegan” R package in the R language and environment was used [39]. To show differences in the species composition of the vegetation under the influence of location (site), the types of material used (control, synthetic or wool) and mechanical stabilisation using ropes (Kemafil vs. no Kemafil), an ordination technique, the Detrended Correspondence Analysis (DCA), was employed. The passive projection of factors (the type of site, type of material and presence/absence of Kemafil ropes) was fit onto the ordination space of DCA. The significance of fit was calculated using a permutation test (999 iterations), and squared correlation coefficients were calculated. In order to examine the effect of the material (ropes, synthetic or wool) and soil on the diversity of species composition, an ordination technique, the Canonical Correspondence Analysis (CCA) was performed using the cca() function [40]. The final model was selected, and any highly correlated variables were excluded according to the Variance Inflation Factor (VIF) using the function vif. cca(). In addition, the ordistep() function was used to determine whether the final model using Akaike’s Information Criterion (AIC) could be reduced. The AIC of the final model, with the AIC for the unconstrained ordination (AIC_0_), was compared to determine whether the constraints improved the quality of the model. The adjusted coefficient of determination for the CCA was obtained using the RsquareAdj() function. Because the classification of the plots was based on the absence/presence of ropes, as well as the presence/absence of wool and synthetic overlap, and because the study plots were situated at four different sites, we wanted to analyse the variance partitioning. In order to show the variance partitioning among the explanatory variables (soil, material and site), Venn diagrams were prepared. For this purpose, the location was treated as the site (Międzyrzecze, Nieboczowy, Lipnik and Wapienica), whereas the soil included all of the chemical compositions of the soil and material, the presence/absence of Kemafil ropes and the presence of wool or synthetic materials.

The differences among the types of material and the types of Kemafil in the biodiversity indices were tested using a modified ANOVA with a permutation test because the data did not fulfil the requirements for parametrical tests. For multiple comparisons, the LSD Fisher test was used. Due to the small number of plots, no interaction between the material and Kemafil was determined.

The effect of the material (synthetic material and Kemafil) on the mean height of the vegetation in the study plots was examined using an analysis of variance (ANOVA) followed by the LSD Fisher test. The normality of distribution and the homogeneity of variance were determined using the Shapiro–Wilk test, the Levene test and the “stats” and “car” packages.

## 3. Results

### 3.1. The Impact of Geotextiles on Species Composition on the Slopes

A list of the species that were identified in the study plots is presented as a supplement in Appendix A. On the slopes, a total of 120 plant species were found. Most of them are meadow, ruderal and segetal species. In Appendix A, the analysis of the soil that covers the ropes is presented. The soil in the plots ranged from slightly acidic to slightly alkaline (4.6 to 7.5 pH), with a low content of organic matter and humus, as well as nutrients (nitrogen, potassium and phosphorus). 

The location of the plots representing different study sites from Wapienica and Lipnik on the left side to Międzyrzecze on the right side of the ordination space is correlated with the first DCA axis. Nieboczowy is at the bottom of the diagram (Figure 3). These differences in location reflect the species turnover among the sites and are significant (Appendix A in in Appendix A). There are no significant differences among the sites that were subjected to different types of material and the presence/absence of ropes.

The CCA demonstrated that, of the 11 environmental variables (soil and type of material), 5 turned out to be significant, and these were phosphorus, pH, humus and the presence of wool and synthetic material (Figure 4 and Appendix A in Appendix A). Wool and synthetic material were correlated with the second axis of the CCA and the ordinate samples across the Lipnik sites. The Wapienica sites were associated with a higher content of wool and humus, whereas the sites in Nieboczowy were mostly confined to wool. In turn, the sites in Międzyrzecze were under the influence of phosphorus (Figure 4a). The species that were more associated with a higher content of wool and humus in the substratum were *Glechoma hederacea*, *Festuca arundinacea*, *Viola tricolor*, *Vicia sativa* and *Scleranthus annuus*, whereas the species that were more associated with a higher content of synthetic material were *Arrhenatherum elatius*, *Lamium amplexicaule*, *Plantago intermedia* and *Myosotis arvensis* (Figure 4b).

According to the Venn diagrams (Figure 4c), the studied variables explained 28% of the species diversity, whereas the highest amount of variance was explained by the soil, followed by the site and the material.

### 3.2. The Influence of Geotextiles on Species Diversity and the Growth of Vegetation

In terms of species diversity, the ANOVA revealed that in the case of the type of material only, there were significant results in the Shannon–Wiener index. A higher value of the H index was found in the control, followed by the synthetic material and wool (Figure 5a). When the plots with and without the Kemafil ropes were compared, there were significant differences in species evenness (Figure 5b).

The ANOVA showed that the type of material (synthetic, wool and control) had an impact on the differences in the mean height of the plants in the study plots. The tallest plants were found in the wool plots, while the shorter ones were found in the synthetic plots, and the shortest plants were found in the control plots (Figure 6a). In Wapienica, Nieboczowy and Międzyrzecze, the wool sites were markedly characterised by a higher mean height of the plants.

The presence of Kemafil ropes had an impact on the mean height of the vegetation in the study plots in all places (Figure 6b).

## 4. Discussion

This study was conducted on slopes at four different sites that had undergone erosion. The sites differed in the substratum (gravel and clay), the type of technology that was used (ropes and fibres) and the material (wool and synthetic). Despite these differences, common patterns of vegetation development were revealed. Both the type of material and the presence of ropes had a higher impact on the quantitative aspects of the vegetation (species diversity and height) than the qualitative parameters (species composition). Thus, this supports our first hypothesis. Regarding Kemafil ropes, it was revealed thaonly the type of material influenced the overall species composition, whilst the presence of ropes did not have such an effect. This is congruent with our second hypothesis. As we assumed, the mechanical stabilisation of the slope caused by the ropes affected the growth of plants but not on species composition.

Despite the similar physiognomy of the vegetation, there were significant differences in floristic composition, which was especially reflected in the case of the Międzyrzecze site (Figure 3a). The use of geotextiles differing in chemical nature and mechanical structure did not influence the vegetation (Figure 3b,c). This was because the experiment was conducted at different sites, and this had a higher impact on the plants that were present and could have caused differences in species composition regardless of the geotextile technology that was used on the slopes. Soil properties, followed by site conditions (plants present in the neighbourhood of the study plot), turned out to be a more important factor affecting species composition than geotextiles (Figure 4c), which is what we expected according to the third hypothesis. Nevertheless, the type of material influenced the vegetation, and it worked across study sites in particular study plots. Some plant species were markedly confined to wool and synthetic material, while other species were more associated with a layer of humus in the soil. The vegetation that grew on the slopes was mostly of a meadow or ruderal character. The typical meadow species that were found were grasses, such as *Arrhenatherum elatius* and *Agrostis capillaris*, and they were more frequent at the sites with a higher content of synthetic material, while dicots, such as *Crepis biennis*, and wetland species, such as *Phalaris arundinacea*, were more confined to wool. Generally, more xerothermic, psammophilous and weedy species, e.g., *Myosotis stricta*, *Lamium amplexicaule* and *Hordeum vulgare*, were found at the “synthetic” sites. We also observed invasive alien plant species, e.g., *Impatiens parviflora* and *Fallopia japonica*, which were more associated with humus in the substratum. Apart from the invasive alien plants, expansive native plant species, which are typical of ruderal habitats, i.e., characterised by intense disturbances and the frequent destruction of vegetation cover, also colonised these sites. The “wool” sites were characterised by a higher content of humus (Figure 4). The apparent correlation between these two variables could be the result of the degradation of wool, which contributes to a higher trophy of soil. Sheep’s wool can act as a fertiliser; this has previously been proven in other studies [41,42,43], which were mainly studies on the fertilising properties of wool on crops. In the present study, geotextiles were mainly used to stabilise the slopes prevent them from eroding. The former studies showed that wool can enhance the growth of grass species and accelerate the growth of an entire vegetation cover [33,34]. The presence of ropes caused the plants to be taller at each site regardless of the material they were made of; however, the wool material caused higher growth than the synthetic material (fourth hypothesis). The ropes had another important function, i.e., stabilising the substratum, which also enabled faster colonisation by the plants. However, species diversity was lower at the sites with a higher content of wool in the soil. We did not expect any differences between the material types, and we especially did not predict that synthetic material could contribute to higher species diversity. During decay, wool provides nitrogen and increases nutrient availability. It has been known that a few species may dominate nutrient uptake in more diverse communities and have effects on species richness [44]. However, in Central Europe, the most commonly reported relationship between species richness and nutrient availability is the hump-shaped curve. Species richness is low at low nutrient levels, increases to a peak at intermediate levels and declines more gradually at high nutrient levels [45]. The obtained result in our study can be explained because an increase in resources (moisture and nutrients) leads to a higher dominance of species, which can result in a decrease in species diversity. This is a reverse situation to the one in highly grazed grasslands, where, through the excretion of faeces, livestock fertilise the soil, thus making grasslands nutrient-rich habitats. Extensive nutrient-rich grazed pastures are seldom characterised by a lower species richness [46]. Usually, they have a higher species diversity. In the cases of the studied slopes, while there was no equivalent disturbance, such as grazing, the former earthworks (the removal of vegetation cover, the making of mounds, the creation of slopes, etc.) are also examples of the disturbance of the vegetation. The analogy to managed grasslands is not fully justified because the origins of the habitats are different. Moreover, the period for colonisation was relatively low, i.e., three to four years after the establishment of the geotextiles, and, therefore, the development of the vegetation had not been completed and stabilised. The control plots represented sites that had not been subjected to the installation of ropes, and they were covered by spontaneously developed vegetation. These sites were mowed as part of the standard earthworks that accompany the maintenance of roads and roadsides by the appropriate road services.

The studied sites and vegetation are not of a very high value from the viewpoint of nature conservation. Thus, the installation of geotextiles can first be applied to increase the rate of greening but not for the maintenance of biodiversity. The post-industrialised sites where they can be refuges for rare and protected plant species are, for instance, waste heaps [47]. In respect of the presence of synanthropic species and the aforementioned origin, i.e., artificially formed slopes, these studied sites resemble waste heaps. The colliery waste tips can be regarded as novel ecosystems sensu Hobbs et al. [48,49]. The patterns of species composition and species diversity at these sites and the occupying plant communities differ from those of natural habitats. The studies on the applicability of geotextiles should be continued in various types of post-industrialised sites determine whether it is possible to encourage or assist their more effective reclamation.

## 5. Conclusions

First of all, the geotextiles efficiently contributed to the growth of the plants at the sites where ropes and fibres were used.

They had a lower impact on the species composition of the vegetation; however, some plant species seemed to be attracted by wool or synthetic material. The sites with the synthetic material were characterised by more xerothermic vegetation, while the sites with wool attracted the plant species typical of meadow vegetation.

The geotextile ropes, especially those containing wool, have the potential for application at many post-industrialised sites because they can efficiently prevent slopes from eroding, stabilise the substratum of artificially formed slopes, store more moisture and provide nutrients.

## Figures and Tables

**Figure 1 materials-15-07957-f001:**
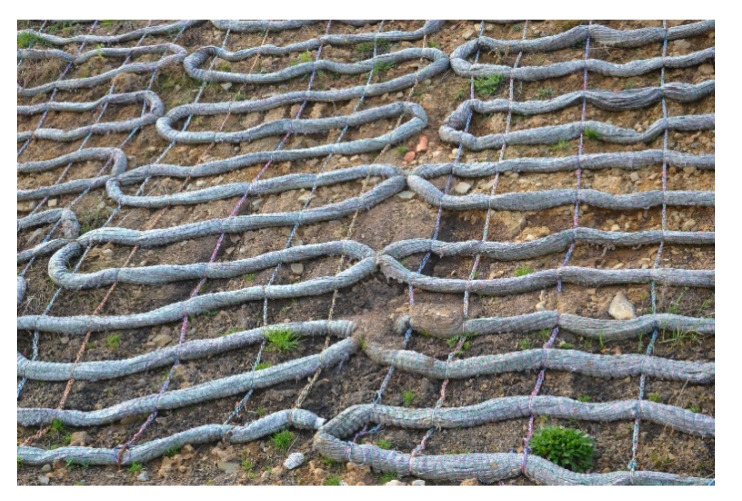
The segments of ropes arranged in a meander-like pattern that were installed on the slope.

**Figure 2 materials-15-07957-f002:**
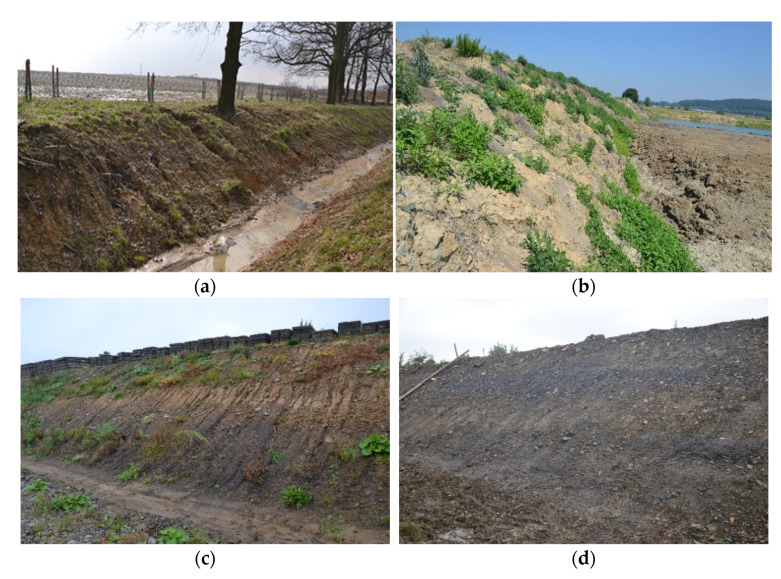
The protected objects before the installation of the ropes. (**a**) Miedzyrzecze; (**b**) Nieboczowy; (**c**) Lipnik; (**d**) Wapienica.

**Figure 3 materials-15-07957-f003:**
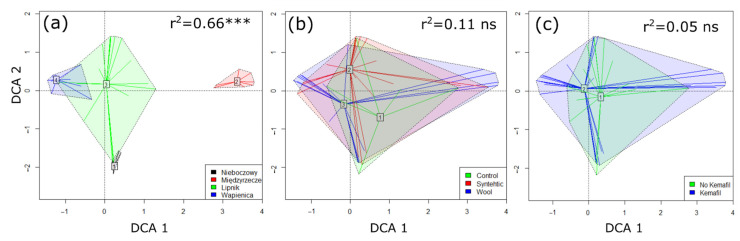
The biplots of Detrended Correspondence Analysis with the passive projection of site (**a**), type of material (**b**) and presence/absence of Kemafil ropes (**c**). *** *p* < 0.001, ns—non-significant.

**Figure 4 materials-15-07957-f004:**
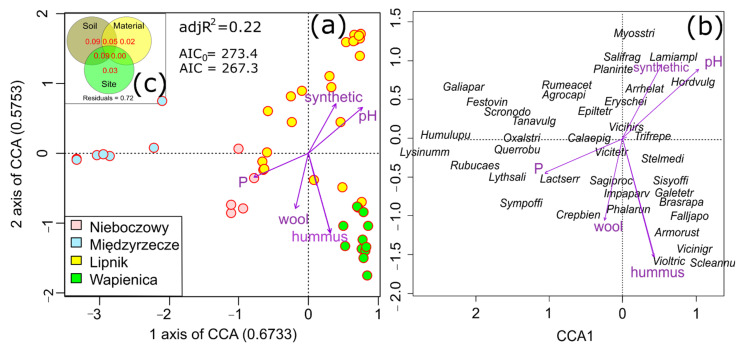
The biplots of the Canonical Correspondence Analysis (CCA): (**a**) diagram showing the study plots within the four sites and the environmental constraints as arrows (only the significant variables are shown); (**b**) diagram showing species; and (**c**) the Venn diagram representing the variance partitioning among the CCA constraints. Explanations: the first four letters denote the genus name and the next four denote the species names.

**Figure 5 materials-15-07957-f005:**
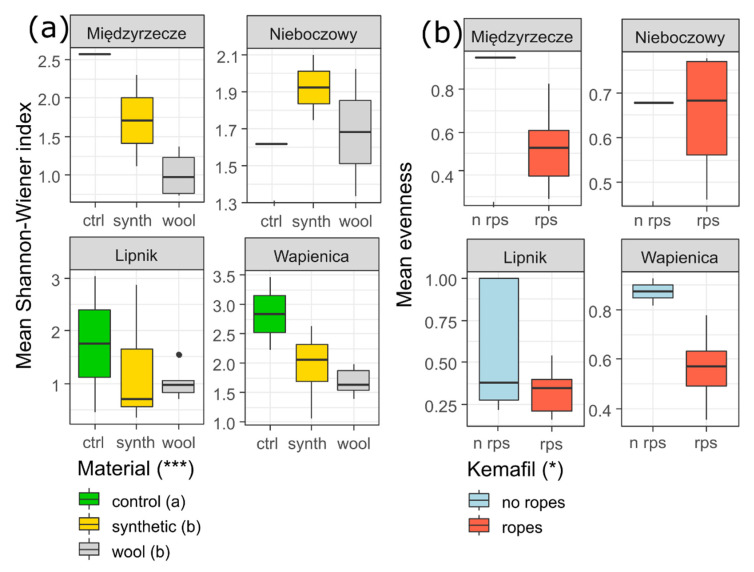
Comparison of the mean value of Shannon–Wiener index of the study plots within and among the sites and among the types of material (**a**) and Kemafil (**b**). Explanation: the ANOVA results refer to the comparison among the material types and between Kemafil types *** *p* < 0.001, * *p* < 0.05. The types of material (indicated by different letters) showed significant differences among the groups of plots according to the post hoc test LSD Fisher test at *p* < 0.05.

**Figure 6 materials-15-07957-f006:**
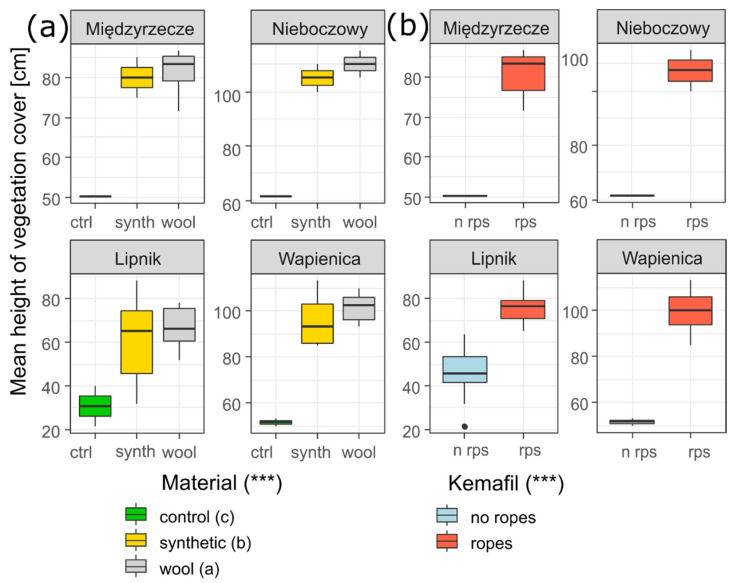
Comparison of the mean value of the height of herb layer in the study plots within and among the sites and among the types of material (**a**) and Kemafil (**b**). Explanation: the ANOVA results refer to a comparison among the material types and between Kemafil types *** *p* < 0.001, The types of material (indicated by different letters) showed significant differences at *p* < 0.05 among the groups of plots according to post hoc test LSD Fisher test.

**Table 1 materials-15-07957-t001:** The objects that were protected using ropes.

Name of the Site	Protected Object	Native Ground	Protective Measure
Miedzyrzecze	Bank of a deep drainage ditch that had been exposed to intense water erosion	Clay	Ropes
Nieboczowy	Unstable slope in an abandoned gravel pit that was prone to local landsliding	Clay + gravel	Ropes
Lipnik	Steep slopes situated between artificially formed flat terraces on a gently sloping hill slope that had been exposed to water erosion	Silt + clay	Ropes + fibres
Wapienica	Newly formed road slope that was prone to shallow gravity landsliding	Clay	Ropes + fibres

**Table 2 materials-15-07957-t002:** The number of study plots at the investigated sites.

Name of the Site	Wool	Synthetic	Control Plots
Miedzyrzecze (n = 7)	4	2	1
Nieboczowy (n = 5)	2	2	1
Lipnik (n = 23)	6	15	2
Wapienica (n = 12)	6	4	2

## Data Availability

Data is available on request from the authors. The data that support thefindings of this study are available from the corresponding author upon reasonable request.

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
