# Peer review of "Multifunctional Geotextiles Produced from Reclaimed Fibres and Their Role in Ecological Engineering"

_materials, 2022, doi:10.3390/ma15227957_

Round 1

Reviewer 1 Report

This was a very interesting paper to read. Although this paper appears to be the latest from a long-running series of investigations, it presents enough new insights to make it of value for anyone interested in the topic. Thus, I have no reservations in recommending its acceptance for publication.

Author Response

Reviewer: This was a very interesting paper to read. Although this paper appears to be the latest from a long-running series of investigations, it presents enough new insights to make it of value for anyone interested in the topic. Thus, I have no reservations in recommending its acceptance for publication.

Response: thank you very much for your positive opinion.

Reviewer 2 Report

Comments to Chmura et al

Summary

The study deals with geotextiles produced from recycled fibers as a means of restraining erosion on sites where human activity has created vulnerable slopes. Wool fibers as well as synthetic fibers are under focus. The authors have conducted field experiments at four different locations protecting selected plots of inclined surface with ropes, made of the recycled fibers, arranged in a meander-like pattern. Two years following the preparation of the slopes, a botanical study gathered data concerning the resulting flora both in the protected plots as well as in unprotected control plots. In addition, there were soil studies providing information on the chemical composition of each plot. A subsequent statistical treatment of the data, employing canonical correspondence analysis, gives information on the effect of the rope and its material on the vegetation.

General comments

The study is topical, and its content is suitable for the Materials journal. Furthermore, the structure of the manuscript is appropriate with the relevant sections included. Moreover, the English language is of good quality and the text is fluent. Likewise, the description of the experiments is comprehensive and instructive for researcher wanting to repeat. As for the statistical analysis, it is quite advanced and would deserve a more thorough presentation. For example, on line 265 we learn that there are twelve environmental variables; in the supporting information, there are eight variables related to the soil and lines 222-223 provide three dummy variables, but I struggle to find the twelfth variable. It would be easier for the reader if there were a table, at least in the supporting information, summarizing the properties of the variables. Correspondingly, the presentation of the results would benefit from a table. The figures are very nice, but it would still be easier to find some specific information in a table. In all, the study suffers to some degree from lack of tangible estimates of the effects on responses like species composition and diversity. Are the differences practically significant or only statistically?

Specific comments

Lines 230-232: This sentence is a bit unclear. What is it that you are decreasing? AIC? Grammatically however, the sentence seems to suggest that you are decreasing the model. If that is indeed your intention, a better choice of words would be to simplify or perhaps reduce the model.

Line 242: What do you mean by residuals?

Line 242: Should it be soil, material and location?

Line 268: Is there a definition or a reference for the concept of vegetational gradient? In general, does the statistical analysis produce any numeric values to assess the species composition?

Lines 278-281: A conjunction, like the word and, joins two parts of a sentence that are grammatically equal. Here however, the second and joins an incomplete clause (no predicate) with a complete clause in the passive voice. You could modify one of the clauses to make the clauses similar or instead replace the word and with the word with, but it may be advisable to split the rather long sentence into shorter ones to improve readability at this important point.

Lines 298-299: What is the difference between, on one hand, the material factor with p<0.001 and material (indicated by letters) with p<0.05. Is the latter an interaction term (material*plot)? Or should it rather be interaction between material and site? I suppose a plot is a subregion of a site. It would be easier to get an overview of the results in the form of a table.

Figure 5: There is no unit on the y-axis.

Lines 312-313: Are the two p<0.05 values just a repetition or are they different?

Line 379: What does slight difference mean? Is it possible to put it on some kind of a scale?

Line 381: Is it the conditions that are xerothermic or just the vegetation?

Line 382: …typical for meadow vegetation.

Author Response

General comments

Reviewer: The study is topical, and its content is suitable for the Materials journal. Furthermore, the structure of the manuscript is appropriate with the relevant sections included. Moreover, the English language is of good quality and the text is fluent.

Response: thank you for this comment and other specific comments that help to improve the text.

R: Likewise, the description of the experiments is comprehensive and instructive for researcher wanting to repeat. As for the statistical analysis, it is quite advanced and would deserve a more thorough presentation. For example, on line 265 we learn that there are twelve environmental variables; in the supporting information, there are eight variables related to the soil and lines 222-223 provide three dummy variables, but I struggle to find the twelfth variable.

Response: Obviously, there are 11 variables. It was mistake, it the present version it was eliminated.

R: It would be easier for the reader if there were a table, at least in the supporting information, summarizing the properties of the variables. Correspondingly, the presentation of the results would benefit from a table. The figures are very nice, but it would still be easier to find some specific information in a table. In all, the study suffers to some degree from lack of tangible estimates of the effects on responses like species composition and diversity. Are the differences practically significant or only statistically?

Response: we partially agree with Reviewer. In supplementary material already two tables with raw data were included: species composition and soil variables. In addition, in the revised version we added a new table where raw data about species diversity and species composition were presented. A new figure in the manuscript, that represent analysis of species composition using ordination e.g. detrended correspondence analysis or DCA, was produced. In plant ecology usually visualization methods (cluster analyses, ordinations) are more frequently presented because they are more beneficial for interpretation of results.

Specific comments

R: Lines 230-232: This sentence is a bit unclear. What is it that you are decreasing? AIC? Grammatically

however, the sentence seems to suggest that you are decreasing the model. If that is indeed your

intention, a better choice of words would be to simplify or perhaps reduce the model.

Response: The sentence was changed. We meant that model will be simplified by reducing number of variables. Now, we wrote that model was reduced, however, in literature also “decreasing model” in the similar context can be found.

R: Line 242: What do you mean by residuals? and Line 242: Should it be soil, material and location?

Response: thank you for the attention to this. Both sentences were changed. Obviously explanatory variables are (soil, material and site or location). The residuals are unexplained variation. It was removed from the description in data analysis. Only we showed it in the figure.

R: Line 268: Is there a definition or a reference for the concept of vegetational gradient? In general,

does the statistical analysis produce any numeric values to assess the species composition?

Response: We modified the sentence and deleted phrase “vegetational gradient”.  In plant ecology we use terms as species turnover, chronosequence, toposequence and rarely - vegetational gradient. It is hard to show single numeric values to assess the species composition. The species composition of vegetation in mathematical way is a matrix of number of species x number of plots. Thus, there are a lot of variables both species and plots. However, in the revised version we conducted DCA analysis and showed difference in species composition among study sites, types of material and presence/absence of Kemafil ropes. These differences were visualized in new figure (Figure 3) but in supplementary material we added some numerical values i.e. position of centroids in ordination space of vegetation.

R: Lines 278-281: A conjunction, like the word and, joins two parts of a sentence that are grammatically

equal. Here however, the second and joins an incomplete clause (no predicate) with a complete

clause in the passive voice. You could modify one of the clauses to make the clauses similar or

instead replace the word and with the word with, but it may be advisable to split the rather long

sentence into shorter ones to improve readability at this important point.

Response: the figure caption was modified. The sentences were rewritten. We divided figure into 3 panels a, b and c and prepared captions to each of them.

R: Lines 298-299: What is the difference between, on one hand, the material factor with p<0.001 and

material (indicated by letters) with p<0.05. Is the latter an interaction term (material*plot)? Or

should it rather be interaction between material and site? I suppose a plot is a subregion of a site. It

would be easier to get an overview of the results in the form of a table.

Response: indeed, the figure caption was misleading. Asterisks showed significance levels of ANOVA test while letters demonstrated significant differences according to post-hoc test i.e. LSD Fisher test. The figure caption was changed.

R: Figure 5: There is no unit on the y-axis.

Response: we added unit [cm] on the y-axis.

R: Lines 312-313: Are the two p<0.05 values just a repetition or are they different?

Response: it was corrected.

R: Line 379: What does slight difference mean? Is it possible to put it on some kind of a scale?

We changed it to ‘marginal’ which is more frequent in the literature.

R: Line 381: Is it the conditions that are xerothermic or just the vegetation?

Response: we agree with suggestion. We changed it into vegetation.

Line 382: …typical for meadow vegetation.

Response: it was revised.

Reviewer 3 Report

The authors should have all the time in mind, starting from the Abstract section until the Conclusion section, to put in value two elements:

1. The originality of the research

2. The study's implications for improving the role of geotextiles' mechanical and chemical properties on the greening of slopes.

 The Abstract sections should be improved. The research ideas could have been more effective through elaborative and concise sentences.

 The literature review section can be improved by adding other theories related to the topic.

 It is necessary to add more details regarding the research data collection, analysis, and interpretation of results.

The results must be interpretive rather than just descriptive and connect the research results with relevant literature citations for validity and reliability.

 The Discussion is not well-presented, as it does not integrate with the research study's results to provide a coherent scholarly argument.

The author should include the critical focus of the study in the Conclusions section.

Unfortunately, the research data does not support the conclusions, which does not indicate a more straightforward path for future studies on the topic.

A follow-up of restated results with supporting literature reviews could make the Conclusion section more effective.

 We suggest that authors pay attention to English.

 Good luck!

Author Response

The authors should have all the time in mind, starting from the Abstract section until the Conclusion section, to put in value two elements:

  1. The originality of the research
  2. The study's implications for improving the role of geotextiles' mechanical and chemical properties on the greening of slopes.

The Abstract sections should be improved. The research ideas could have been more effective through elaborative and concise sentences.

Response: We provided new sentences and some of old sentences were rewritten. The originality of research was highlighted.

The literature review section can be improved by adding other theories related to the topic.

Response: In discussion we added some new references that are related to theories about relationship between nutrient availability and species richness, species dominance as well. It can help to interpret the obtained results particularly regarding role of wool.

It is necessary to add more details regarding the research data collection, analysis, and interpretation of results.

Response: we added two new tables in supplementary material and one figure in the text.

The results must be interpretive rather than just descriptive and connect the research results with relevant literature citations for validity and reliability.

Response: we added hypotheses at the end of introduction. Results follow them.

The Discussion is not well-presented, as it does not integrate with the research study's results to provide a coherent scholarly argument.

Response: The fragments of discussion were rewritten. The particular sections refer to results that were supposed to support or reject hypotheses.

The author should include the critical focus of the study in the Conclusions section.

Response: In the original version already it was mentioned.

Unfortunately, the research data does not support the conclusions, which does not indicate a more straightforward path for future studies on the topic.

Response: Conclusions were rewritten.

A follow-up of restated results with supporting literature reviews could make the Conclusion section more effective.

Response: New references were added.

 We suggest that authors pay attention to English.

Response: the original version of the paper was proofread by native speaker.

Round 2

Reviewer 3 Report

Good luck!